# Proanthocyanidins and Flavan-3-Ols in the Prevention and Treatment of Periodontitis—Antibacterial Effects

**DOI:** 10.3390/nu13010165

**Published:** 2021-01-07

**Authors:** Izabela Nawrot-Hadzik, Adam Matkowski, Jakub Hadzik, Barbara Dobrowolska-Czopor, Cyprian Olchowy, Marzena Dominiak, Paweł Kubasiewicz-Ross

**Affiliations:** 1Department of Pharmaceutical Biology and Botany, Wroclaw Medical University, 50556 Wroclaw, Poland; izabela.nawrot-hadzik@umed.wroc.pl; 2Department of Dental Surgery, Wroclaw Medical University, 50425 Wroclaw, Poland; jakub.hadzik@umed.wroc.pl (J.H.); cyprian.olchowy@umed.wroc.pl (C.O.); marzena.dominiak@umed.wroc.pl (M.D.); pawel.kubasiewicz-ross@umed.wroc.pl (P.K.-R.); 3Department of Clinical Nursing, School of Health Sciences, Wroclaw Medical University, 50556 Wroclaw, Poland; barbara.dobrowolska-czopor@umed.wroc.pl

**Keywords:** condensed tannins, proanthocyanidins, flavan-3-ols, periodontitis, gingivitis, gum disease, cranberry, tea, polyphenols, natural compounds, natural substances

## Abstract

Flavan-3-ols and their oligomeric forms called proanthocyanidins are polyphenolic compounds occurring in several foodstuffs and in many medicinal herbs. Their consumption is associated with numerous health benefits. They exhibit antioxidant, anti-inflammatory, cytoprotective, as well as antimicrobial activity. The latter property is important in the prevention and treatment of periodontal diseases. Periodontitis is a multifactorial polymicrobial infection characterized by a destructive inflammatory process affecting the periodontium. Using non-toxic and efficient natural products such as flavanol derivatives can significantly contribute to alleviating periodontitis symptoms and preventing the disease’s progress. Therefore, a comprehensive systematic review of proanthocyanidins and flavan-3-ols in the prevention and treatment of periodontitis was performed. The present paper reviews the direct antibacterial effects of these compounds against periodontic pathogens. The immunomodulatory effects, including animal and clinical studies, are included in a separate, parallel article. There is significant evidence supporting the importance of the antibacterial action exerted by proanthocyanidins from edible fruits, tea, and medicinal herbs in the inhibition of periodontitis-causing pathogens.

## 1. Introduction

Periodontitis is a multifactorial polymicrobial infection characterized by a destructive inflammatory process affecting the periodontium, which comprises a set of teeth supporting structures: gingiva, cementum, periodontal ligament, and alveolar bone. Approximately 5 to 15% of the world population is affected by severe forms of the disease, which, if left untreated, may result in tooth loss and systemic complications [1,2,3]. In the last 30 years, the classification of periodontitis has been modified in an attempt to align it with emerging scientific evidence. Based on the pathophysiology state-of-the-art, a consensus was reached during the World Workshop for Periodontology in 2017, on identifying three periodontitis forms: necrotizing periodontitis, periodontitis as a manifestation of systemic disease, and the previously recognized as “chronic” or “aggressive” forms of the disease, now grouped under a single category—“periodontitis” [4]. The most current concept of the etiopathology of periodontitis involves the co-existence of dental plaque and host immune-inflammatory response. Socransky and Haffajee [5] divided the periopathogens involved in periodontitis into six clusters: red, orange, yellow, green, blue, and purple. The first to colonize the surface of the teeth are “purple” and “yellow” complexes comprised mostly by *Actinomyces* species and *Streptococci* including *S. sanguinis* and *S. oralis*. The next complex, involved in periodontitis progression, includes the bacteria contributing to the primary changes in the host: *Aggregatibacter actinomycetemcomitans*, *Campylobacter concisus*, *Capnocytophaga* spp., and *Eikenella corrodens*. The “orange” cluster includes the “bridging species” capable of using and secreting nutrients in the biofilm, in addition to expressing cell surface molecules facilitating binding to early colonizers and the individuals of the red complex. This cluster includes the following species: *Prevotella* spp., *Micromonas micros*, *Fusobacterium* spp., *Eubacterium* spp., and *Streptococcus constellatus*. Finally, *Porphyromonas gingivalis* and *Treponema denticola* in addition to *Tannerella forsythia* are responsible for the further progression of periodontitis are referred to as the red cluster [5].

The presence of bacteria is necessary, but insufficient to cause periodontal disease. The exposure to bacteria must be connected with the individual’s susceptibility. However, the individual’s susceptibility to bacteria is dependent on genetic factors. The structure of the periodontium, which itself is a barrier to periodontal pathogens, is also genetically predetermined. Moreover, the individual’s response to the inflammation of periodontal tissue and its medical course are under the influence of a number of environmental factors [6]. Periodontal bacteria lead to the mobilization of innate immune response-related signaling mediators (e.g., IL-1, IL-6, TNF-α), as well adaptive immunity mechanisms (expression of Th1, Th2, Th17, and Tregs). Beside periopathogens, the host response plays a leading role in pathogenesis of periodontitis. Specifically, the overproduction of inflammatory mediators such as pro-inflammatory cytokines, prostanoids and matrix metalloproteinases (MMPs) by resident and immune cells can modulate the progression and severity of periodontitis.

The high antimicrobial and immunomodulatory activities of proanthocyanidins make them an interesting class of phytonutrients for the prevention and treatment of periodontal diseases [7,8]. An additional benefit is their natural dentin cross-linker activity and inhibition of MMPs, which may be helpful in adhesive dentistry [9]. The bioactivity of proanthocyanidins arises from their unique chemical structure [10]. Proanthocyanidins are highly hydroxylated structures built from flavan-3-ol blocks, forming oligomeric structures of various numbers of units (from two to many). Mostly, the flavan-3-ol units are catechin (C), epicatechin (EC), or their substituted derivatives connected through the C4–C8 or C6 bonds (B-type). The term condensed tannins refers to their ability to form insoluble complexes with carbohydrates and proteins [9]. According to the number of hydroxyl substitutions in the B ring, proanthocyanidins can be categorized as propelargonidin (one hydroxyl substitution), procyanidin (two hydroxyl substitutions), and prodelphinidin (three hydroxyl substitutions) (Figure 1). 

B-type proanthocyanidins are found in a variety of food sources, such as grapes, red wine, chocolate, black chokeberry, as well as in many plants used in traditional medicine like the rhizome of *Reynoutria japonica* Houtt. (synonym *Polygonum cuspidatum*) [11,12] or *Sanguisorba officinalis* L. [13] and many others [14]. A-type polymers isolated from cranberry are less common. They are distinguished by having at least one intermolecular bond between the O7 and C2 atoms in addition to the linkage by carbon-carbon bonds [15] (Figure 2). 

The unique configuration of a particular proanthocyanidin structure can influence its biological activity in periodontitis. Some simple (not condensed) molecules such as catechin, epicatechin, and their derivatives from tea are also helpful in periodontal diseases [16]. The aim of this review was to verify and discuss evidence that proanthocyanidins and flavan-3-ols are beneficial in the prevention and treatment of periodontitis. The present review is a comprehensive systematic review of proanthocyanidins and flavan-3-ols in the prevention and treatment of periodontitis prepared in compliance with the PRISMA guidelines [17]. Due to the wide scope of the topic, the present paper focuses on the direct effects of procyanidins on periopathogens. The immunomodulatory, including animal and clinical, evidence will be covered in the second part of this review series [18].

## 2. Methods 

### 2.1. Search Strategy

This systematic review adhered to the Preferred Reporting Items for Systematic Reviews and MetaAnalysis (PRISMA) guidelines [17]. An electronic database search was conducted using PubMed, Scopus, and Web of Science (as of 23 December 2020). The search terms included all combinations of the following key words: periodontitis OR periodontal diseases OR gingivitis OR gingival diseases AND proanthocyanidins OR condensed tannins OR flavan-3-ols OR catechin OR epicatechin AND anti-bacterial OR antiadhesive OR anti-inflammatory, respectively. All titles with abstracts were imported into a citation manager program “Mendeley” (Elsevier-Mendeley Ltd., London, UK), and all duplicates were removed. Bibliographies of imported articles were also screened for other relevant studies. Two investigators (I.N.-H. and P.K.-R.) independently reviewed the titles and abstracts of the imported references to determine whether they met the inclusion and exclusion criteria. Disagreements were resolved via consensus and by a third investigator (K.H.). 

### 2.2. Inclusion Criteria

The inclusion criteria were as follows: (a) all relevant studies reporting the influence of proanthocyanidins or flavan-3-ols on the growth, colony formation, and metabolic activity of periopathogens and studies reporting the possible inhibition of periopathogens’ adhesion to potential oral mucosa cells; (b) all relevant in vitro studies reporting the immunomodulatory effects of proanthocyanidins or flavan-3-ols on host cells or periodontal tissue treated with exotoxins from periopathogens; (c) all relevant in vivo studies reporting the influence of proanthocyanidins or flavan-3-ols on periodontitis in animal models; (d) clinical trials studying the influence of proanthocyanidins or flavan-3-ols on periodontitis. Only studies published in the English language were taken into consideration. All included articles were critically read and analyzed. If there were any uncertainties regarding the quality of a study not filtered out during the preliminary assessment, it is described in the manuscript.

### 2.3. Exclusion Criteria

Review articles and prospective or in-silico only studies were excluded from the present study. 

Further, we excluded the following types of experimental papers: studies on poorly characterized plant extracts or extracts without the confirmed presence of proanthocyanidins or flavan-3-ols;studies involving oral pathogens, but not specific periodontal pathogens;the application of proanthocyanidins or flavan-3-ols in combination with other antimicrobial pharmaceuticals, e.g., chlorhexidine or antibiotics.

### 2.4. Data Organization

The authors, year of publication, type of study, type of compounds, plant source of compounds, compound concentration, type of bacteria, type of cells and tissues, methods, and principal findings of each study were noted in a standard document. The studies were divided into four groups following the inclusion criteria: (1) studies reporting the antibacterial effects on periopathogens and inhibiting bacterial proteolytic enzymes by proanthocyanidins or flavan-3-ols; (2) in vitro studies reporting the immunomodulatory effects of proanthocyanidins or flavan-3-ols on host cells and tissues; (3) in vivo studies reporting the influence of proanthocyanidins or flavan-3-ols on periodontitis in animal models; (4) clinical studies.

## 3. Results and Discussion

After duplicate removal, one-hundred thirty-four articles were further screened by the title and abstracts (Figure 3).

Finally, sixty-five studies met the inclusion criteria: 31 of these in vitro studies reported the influence of proanthocyanidins or flavan-3-ols on the growth, colony formation, and metabolic activity of potential periopathogens and the inhibition of periopathogens adhesion to oral mucosa cells; 36 in vitro studies reported the action of proanthocyanidins or flavan-3-ols in the immunological response of the periodontal tissues; 11 in vivo studies reported the influence proanthocyanidins or flavan-3-ols on periodontitis in animal models; and three controlled clinical trials reported the application of proanthocyanidins or flavan-3-ols in periodontitis. In some references (*n* = 16), the in vitro antibacterial tests were performed concurrently with investigations into the immunomodulatory effects in host cells or in animal models. The results of these studies pertaining to the inflammatory processes are reviewed in our other paper, separately [18].

### Antibacterial Effects of Proanthocyanidins or Flavan-3-Ols on Periopathogens

An overview of the antibacterial effects of the proanthocyanidins or flavan-3-ols is presented in Table 1. 

The majority of the studies (22) reported effects against *Porphyromonas gingivalis*. Other pathogens were included in much fewer reports. Five studies included *Aggregatibacter actinomycetemcomitans*, 3 studies *Fusobacterium nucleatum*, 2 studies against *Prevotella intermedia* or *Treponema denticola*, only single studies using *Tannerella forsythia*, *Eikenella corrodens*, and *Peptostreptococcus micros* cells, as well as one study against oral polymicrobial biofilms. The influence of proanthocyanidins or flavan-3-ols on bacterial enzymatic activity was also reported by most of these studies. 

The Gram-negative anaerobic rod *P. gingivalis*, the most studied bacterium, can adhere to epithelial cells of the gingival mucosa and endothelial cells using fimbriae OMPs (outer membrane proteins). It can also produce a series of high virulence factors like proteases (e.g., collagenase), hemolysins, endotoxins, fatty acids, ammonia, hydrogen sulfide, indole, and others, which are important for adherence, colonization, and nutrient acquisition and which affect the host immune response [14,15]. Some of them, like lipopolysaccharide (LPS), binds the toll-like receptors (TLRs) (expressed in various immune cells, such as neutrophils, macrophages, and dendritic cells), activates inflammatory signaling pathways, promotes the secretion of pro-inflammatory cytokines, nitric oxide (NO), and eicosanoids, and finally, causes the symptoms of inflammation [21]. However, it is supposed that the most important virulence factors are cysteine proteases, the arginine-specific (RgpA and RgpB) and lysine-specific (Kgp) gingipains, which are attributed to 85% of the total proteolytic activity of *P. gingivalis* [49]. Moreover, they are the most potent adhesins of *P. gingivalis.* They are located on the surface of *P. gingivalis* cells from which subfractions are secreted into the extracellular fluid [30]. Gingipains execute pathological actions due to their reactivity with a broad-range of targets, such as cytokines. They are essential for tissue degradation and may contribute to the penetration of this bacterium into the periodontium [50].

Fractions rich in proanthocyanidins (PACs, often named APACs because of the A-type bond) from cranberry fruits (*Vaccinium macrocarpon*) are among the most studied natural substances against periopathogens. Proanthocyanidins isolated from cranberries are mainly composed of epicatechin subunits with at least one A-type bond (intermolecular bond between O7 and C2 in addition to the carbon-carbon bond). According to the collected literature data (Table 1), cranberry PACs can inhibit *P. gingivalis* attachment to the periodontal tissue and reduce bacterial biofilm formation, collagenase activity, and invasion by neutralizing periodontopathogen proteinases and cytotoxicity, but they do not interfere with the growth of *P. gingivalis* [20,21,34,36,39,42]. La et al. [39], in addition to the above activities, showed that A-type cranberry proanthocyanidins inhibited the adherence of *P. gingivalis* to Matrigel-coated polystyrene surfaces and inhibited type I collagen degradation by extracellular proteinases produced by *P. gingivalis* in a dose-dependent manner. Despite that, PACs did not influence *P*. *gingivalis* growth by themselves. Ikai et al. [35] showed that bactericidal activity against *P. gingivalis* and *S. mutans* of the hydrogen peroxide photolysis system was augmented in the presence of 2–8 mg/mL commercial grapeseed proanthocyanidins. A putative mechanism of action could involve additional H_2_O_2_ generation up to 1 mM by irradiated PACs dissolved in an aqueous buffer (PBS), as was demonstrated using an EPR (electron paramagnetic resonance) detection. Feldman and Grenier [36] also proved that the bactericidal effect of cranberry proanthocyanidins could be improved in the presence of another polyphenol. They observed that when PAC and licochalcone A were used in combination, *P. gingivalis* growth was inhibited in a synergistic manner. 

Bodet at al. [42] presented the effect of the PAC fraction from cranberry juice on the proteolytic activities of *P. gingivalis* and two other periopathogens—*Tannerella forsythia* and *Treponema denticola*, belonging to the “red cluster”, the most responsible for the progression of periodontitis. Both *T. forsythia* and *T. denticola* produced proteases that contributed to bacterial virulence in multiple ways. The proteolytic activities can act through degradation of host periodontal tissues, activation of the host’s degrading enzymes, modifying host cell proteins, and cleaving components involved in innate (cytokines/chemokines, complement factors) and adaptive(immunoglobulins) immunity [51]. Bodet at al. [42] noticed that the PAC fraction dose-dependently (10–150 μg/mL) inhibited the proteolytic activities of *P. gingivalis* (Arg-gingipain, Lys-gingipain, dipeptidyl peptidase IV (DPP IV)), *T. forsythia* (trypsin-like proteinase) and *T. denticola* (chymotrypsin-like proteinase); however, the trypsin-like activity of *T. forsythia* was only slightly sensitive to PACs. Moreover, the proanthocyanidin fraction significantly reduced the collagenase activity of *P. gingivalis* and the capability of *P. gingivalis* to degrade transferrin. The degradation of type I collagen and transferrin by *P. gingivalis* was completely or almost completely inhibited by 100 μg/mL and 150 μg/mL of non-dialyzable material (NDM), respectively [42].

Proanthocyanidins from other sources than cranberry and with different structures were also studied. De Oliveira Caleare et al. [27] studied a 70% acetone extract from *Limonium brasiliense* rhizomes (LBEs), rich in proanthocyanidins, EGCG, and gallic acid. LBEs contained a high amount of untypical double linked proanthocyanidins—samarangenins A and B (Figure 4). 

LBEs at 100 μg/mL reduced the adhesion of *P. gingivalis* to the human epithelial KB cells by about 80% and at 20 μg/mL reduced the proteolytic activity of the arginine-specific Rgp gingipain by about 75%. LBEs at ≤100 µg/mL had no cytotoxicity against the bacteria and did not influence the cell physiology of human epithelial KB cells. Findings from the study of Löhr et al. [37] showed that 50% ethanol extract from *Myrothamnus flabellifolia* (MF), rich in flavan-3-ols and oligomeric B-type proanthocyanidins, dose-dependently inhibited *P. gingivalis* attachment to or invasion of KB cells; however, bacterial growth was not influenced. Moreover, MF extract at 50 μg/mL reduced Arg-gingipain by 70–80% and also inhibited Lys-gingipain, but to a lesser extent. Schmuch at al. [30] carried out an extensive study about the influence of a proanthocyanidin-enriched extract from the aerial parts of *Rumex acetosa* (sorrel) (RA1) and isolated compounds (details in Table 1) against the adhesion of *P. gingivalis.* It was revealed that RA1 (5 to 15 μg/mL) reduced *P. gingivalis* adhesion to KB cells in a dose-dependent manner to about 90% at 15 μg/mL. Galloylated proanthocyanidins were confirmed to be responsible for this antiadhesive effect with procyanidin B2-di-gallate being the lead compound. A trigalloylated trimeric procyanidin (epicatechin-3-O-gallate-(4β→8)-epicatechin-3-O-gallate-(4β→8)-epicatechin-3-O-gallate) was even more active than procyanidin B2-di-gallate, but it was a minor compound in the RA1 fraction. The compounds not esterified with gallic acid—flavan-3-ols (epicatechin, catechin, epigallocatechin and gallocatechin) and oligomeric proanthocyanidins (procyanidin B2 and epicatechin-(4β→8)-epicatechin-(4β→8)-catechin)—were inactive similar to the quercetin-3-O-glucuronide present in large amounts in the RA1. Interestingly, a non-galloylated, mixed A/B-type proanthocyanidin also present in the RA1 fraction—cinnamtannin—reduced *P. gingivalis* adhesion to KB cells, but only moderately. Moreover, RA1 and the galloylated proanthocyanidins strongly inhibited Arg-gingipain. The inhibition force increased with the degree of polymerization: the galloylated trimer had higher activity than dimers and monomers, which were barely active. No differences were observed between the dimeric galloylated 4→8-linked and the 4→6-linked proanthocyanidins. Moreover, monogalloylation in the lower building block seemed to be sufficient for activity, while di-galloylation did not seem to be necessary. Again, a mixed A/B-type proanthocyanidin—cinnamtannin—was less active. Contrary to Arg-gingipain, Lys-gingipain was hardly influenced by RA1 and its constituents. Lys-gingipain activity was only influenced to a minor extent by the di- and tri-meric galloylated procyanidins. RA1 also inhibited *P. gingivalis*-induced hemagglutination, but did not influence the gene expression of *rgpA* (for Arg-gingipain), *kgp* (for Lys-gingipain), and *fimA* (for fimbrillin). In silico docking studies indicated that procyanidin B2-di-gallate interacts with the active side of Arg-gingipain and hemagglutinin from *P. gingivalis* and that the galloylation of the molecule seems to be responsible for the fixation of the ligand to the protein. Expectedly, the total amount of H-bond donors was an important factor, indicating a tannin-like effect and therefore suggesting an unspecific interaction with the hemagglutination domain [30]. 

The results of Okamoto et al. [44] are consistent with the above report. Only tea catechin derivatives containing the galloyl moiety inhibited the Arg-gingipain and to a lesser extent the Lys-gingipain of *P. gingivalis.*

Amel Ben Lagha et al. [24] showed that epigallocatechin gallate (EGCG) and the extract from green tea inhibited both Arg-gingipain and Lys-gingipain activities. However, the green tea extract was the stronger inhibitor. Both similarly inhibited the type I collagen degradation by *P. gingivalis*. Collagen degradation by *P. gingivalis* is mainly related to its Arg-gingipain activity [52]. In the same study, EGCG and the green tea extract protected the epithelial barrier against damage caused by *P. gingivalis* and prevented bacterial penetration through a keratinocyte monolayer. It was linked with the ability of these substances to enhance gingival epithelium barrier function and with their influence on gingipains [24]. A similar effect protecting the epithelial barrier against *P. gingivalis*-mediated damage was reported earlier for black tea theaflavins (Figure 5) [26]. 

Moreover, theaflavins dose-dependently inhibited the gene expression of major virulence factors of *P. gingivalis*, such as: *fimA*, *hagA*, involved in colonization, and *rgpA* and *kgp*, responsible for host tissue destruction. Fournier-Larente et al. [28] showed the same activity for green tea extract and EGCG, extending the studied genes by *hagB* and *htrA* (involved in stress response), as well as *hem* (involved in heme acquisition). Sakanaka and Okada showed that green tea polyphenols reduce the ability of *P. gingivalis* to produce toxic end metabolites (n-butyric, propionic acid, and phenylacetic acid), which injure periodontal cells and disturb host cell activity [45]. The inhibitory effect on the production of toxic end metabolites can be attributed to the presence of the galloyl moiety in the catechins. The growth of *P. gingivalis* was inhibited by EGCG, but only at a high concentration: 0.125–0.5 mg/mL [45]. In the earlier study, Sakanaka et al. [48] showed that the same extract and galloylated flavan-3-ols ((−) epicatechin gallate, EGCG, (−) gallocatechin gallate) significantly inhibited *P. gingivalis* adherence onto the buccal epithelial cells. This activity was attributed to the presence of the galloyl moiety. EGCG also completely inhibited the growth of three strains of *P. gingivalis* at concentrations of 250 and 500 μg/mL, whereas the MIC for other polyphenols was 1000 μg/mL. Similarly, in the study by Hirasawa et al. [47], the MICs of green catechins from Japanese green tea (unspecified catechins) for *P. gingivalis*, *P. intermedia*, and *P. nigrescens* were 1.0 mg/mL. Asahi et al. [33] demonstrated that EGCG at high concentrations (500 μg/mL or 5 mg/mL) disrupted established *P. gingivalis* biofilms, by the destruction of the bacterial cell membrane. Moreover, EGCG at sub-MIC levels (10 μg/mL and 100 μg/mL) inhibited *P. gingivalis* biofilm formation. EGCG at 10 μg/mL efficiently inhibited biofilm formation without affecting the growth rate. At sub-MIC, EGCG did not damage the cell membrane of *P. gingivalis.* Hence, the inhibition of *P. gingivalis* biofilm formation is likely based on a mechanism distinct from that responsible for its bactericidal activity at high concentrations. Matsunaga et al. [38] also observed remarkable inhibition of *E. corrodens* biofilm formation by sub-MIC levels of catechins with the pyrogallol-type B-ring and/or the galloyl group. As the research showed, this may by through the interference with the AI-2-mediated QS system in *E. corrodens.* Gao et al. [21] studied the buds of *Castanopsis lamontii* Hance water extract (CLE), rich in epicatechin and procyanidin B2, confirming the inhibition of *P. gingivalis* growth by flavan-3-ols or proanthocyanidins only at high concentrations. 

Other plant materials rich in procyanidins, studied against *P. gingivalis*, are *Pelargonium sidoides* roots [25], *Ulmus macrocarpa* bark [46], and apples and hop bracts [43]. Savickiene et al. [25] demonstrated that *Pelargonium sidoides* root extract (PSRE), rich in monomeric flavan-3-ols (e.g., epigallocatechin, catechin, EGCG) with a minor contribution of proanthocyanidins, as well as the proanthocyanidin-enriched fraction obtained from PSRE (PACN), significantly reduced the viability of *P. gingivalis* and the non-pathogenic commensal *Streptococcus salivarius*, starting at a concentration of 0.05 mg/mL. However, the PACN fraction was partially selective against *P. gingivalis*. Song et al. [46] studied a partially purified extract from the bark of *Ulmus macrocarpa*, defined as elm extract (containing 20% of procyanidins) and its active ingredient, a mix of proanthocyanidin oligomers (composed of three to 12 monomers, an average molecular weight of 1518) for a possible inhibitory effect against proteases—trypsin-like enzymes from *P. gingivalis* and *Treponema denticola.* Both fractions inhibited the proteases of these pathogens, but the proanthocyanidin oligomer mixture inhibited them more effectively than the elm extract. The trypsin-like activity of *T. denticola* was slightly more susceptible to these inhibitory effects than *P. gingivalis*. Inaba et al. [43] studied fractions rich in proanthocyanidins from immature apples (*Malus* sp.) and hop bracts (*Humulus* sp. from Japan). The apple fraction (AP) and a more purified fraction called apple condensed tannins (ACTs) are oligomeric, whereas the hop bract polyphenol fraction (HBP) and its high molecular weight fraction (HMW-HBP) are polymeric proanthocyanidins. The studied fractions at very low concentrations of 1–10 μg/mL significantly protected periodontal ligament (PDL) cells’ viability from the effect of *P. gingivalis* infection, although EGCG and LMW-HBT (low molecular weight fraction of HBT) showed lower effects than the others (AP, ACT, HBP, HMW-HBP). All fractions inhibited the proteolytic activities of Rgp and Kgp in a dose-dependent manner, with AP, ACT, and HBP being more effective toward Kgp. Moreover, AP, ACT, HBP, and HMW-HBP significantly protected enamel matrix derivative (EMD)-stimulated PDL cells from *P. gingivalis*, suggesting a potential benefit of using proanthocyanidins to enhance periodontal tissue regeneration in response to EMD. In contrast, EGCG and LMW-HBP were inactive, suggesting that higher polymerized procyanidins are responsible for the above effect.

Ben Lagha et al. [23] reported that proanthocyanidins isolated from highbush blueberry (*Vaccinium corymbosum*) reduced the growth of *Aggregatibacter actinomycetemcomitans* and prevented biofilm formation at sub-inhibitory concentrations. This effect was linked to the ability of PACs to damage the bacterial cell membrane. The application of PACs on pre-formed biofilms resulted in a loss of bacterial viability. Moreover, PACs significantly reduced LtxA cytotoxicity towards macrophage-like cells and protected the oral keratinocytes’ barrier integrity from damage caused by *A. actinomycetemcomitans*. Another study of the same group [22] tested the influence of cranberry PACs from cranberries on the mRNA expression of *A. actinomycetemcomitans* leukotoxin encoding genes. PAC (60 µg/mL) treatment downregulated the mRNA level of ltxB by 65.3% and 88.7% and of ltxC by 94.4% and 86.1% in the Y4 and JP2 strains, respectively. LtxB encodes the components required for the transport of LtxA to the *A. actinomycetemcomitans* outer membrane, and ltxC encodes components involved in posttranslational acylation. 

The above-mentioned *Pelargonium sidoides* root extract (PSRE) and proanthocyanidin fraction (PACN) were also active against *A. actinomycetemcomitans* [20]. PSRE and PACN at 80 µg/mL significantly reduced bacterial metabolic activity in comparison to the untreated control, whereas PACN was more effective than PSRE. Moreover, PSRE and PACN protected human gingival fibroblast from *A. actinomycetemcomitans* infection through lowering bacteria proliferation and prevented LPS-induced necrosis.

Hata et al. [17] reported the antimicrobial activity of epigallocatechin gallate (EGCG) against *A. actinomycetemcomitans* at >0.5 mg/mL. However, EGCG also precipitated several salivary proteins including α-amylase, thus inhibiting the enzymatic activity. On the other hand, α-amylase reduced the antimicrobial activity of EGCG. It was suggested that EGCG–salivary protein interactions may have both protective and detrimental effects on oral health. This should certainly be considered when assessing the effects of EGCG on the oral cavity and probably also applies to many proanthocyanidins with protein-binding activity.

## 4. Conclusions

Among the reviewed in vitro studies, thirty-one reported an influence of proanthocyanidins or flavan-3-ols on periopathogens, mainly *Porphyromonas gingivalis* (22 studies). Much fewer studies concerned other oral pathogens: *Aggregatibacter actinomycetemcomitans*, *Fusobacterium nucleatum*, *Prevotella intermedia*, *Treponema denticola*, *Tannerella forsythia*, *Eikenella corrodens*, *Peptostreptococcus micros.* Both proanthocyanidins and simple flavan-3-ols affected the attachment of periopathogens, mostly *P. gingivalis*, to the periodontal tissue, depending on their chemical structure. The antiadhesive effect was attributed to the presence of the galloyl moiety in the B-type proanthocyanidins or flavan-3-ols (e.g., from tea) and an A-type linkage in the case of A-type proanthocyanidins from cranberry. Similarly, this structural pattern was also important for other activities, such as the reduction of bacterial biofilm formation, collagenase activity, as well as in neutralizing periodontopathogen proteinases’ activity and cytotoxicity. The above-mentioned activities were manifested at low, micromolar concentrations, at which they only slightly interfered with periopathogen growth. However, inhibition often occurred at higher concentrations. 

Using flavanol derivatives at their non-toxic, but active concentrations can significantly contribute to alleviating periodontitis symptoms and preventing the disease’s progress. However, considering the usage of these compounds in the prevention and treatment of periodontitis, their interaction with saliva proteins should be taken into account, because they may alter their level of antimicrobial activity, which needs to be looked at in the future.

## Figures and Tables

**Figure 1 nutrients-13-00165-f001:**
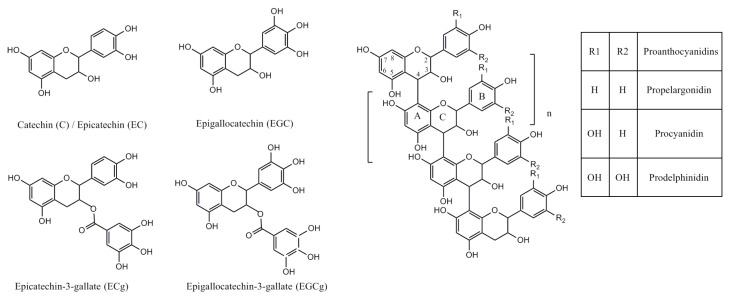
Structure of flavan-3-ols and proanthocyanidins.

**Figure 2 nutrients-13-00165-f002:**
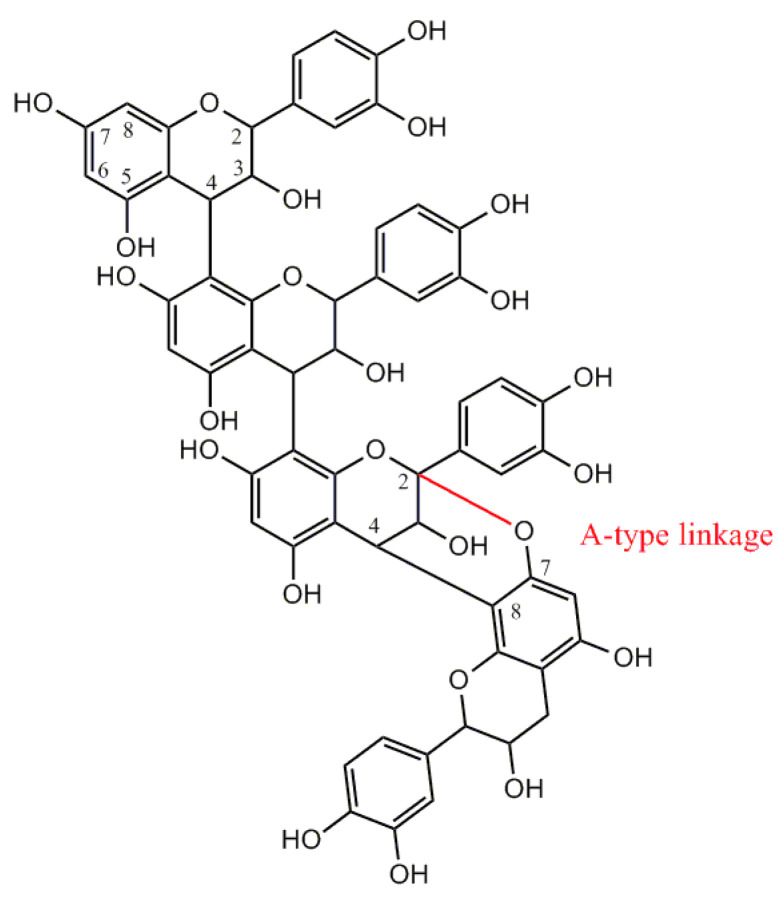
Structure of cranberry proanthocyanidins with A-linkage.

**Figure 3 nutrients-13-00165-f003:**
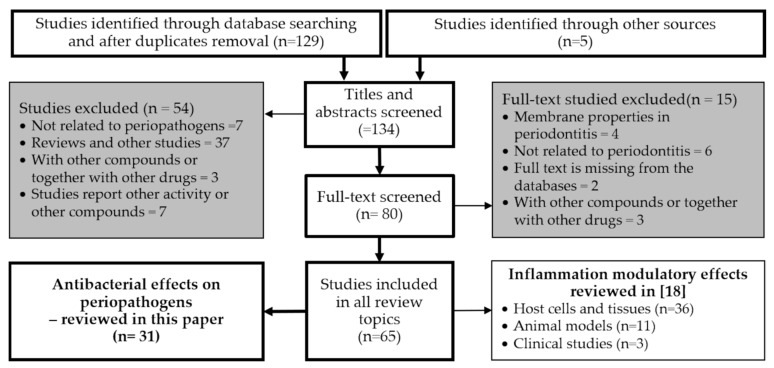
Flowchart of the article search strategy, exclusion criteria, study selection, and data management process. Of all 65 considered references, thirty-one are reviewed in this paper and 50 in [18], of which 16 references are included in both reviews.

**Figure 4 nutrients-13-00165-f004:**
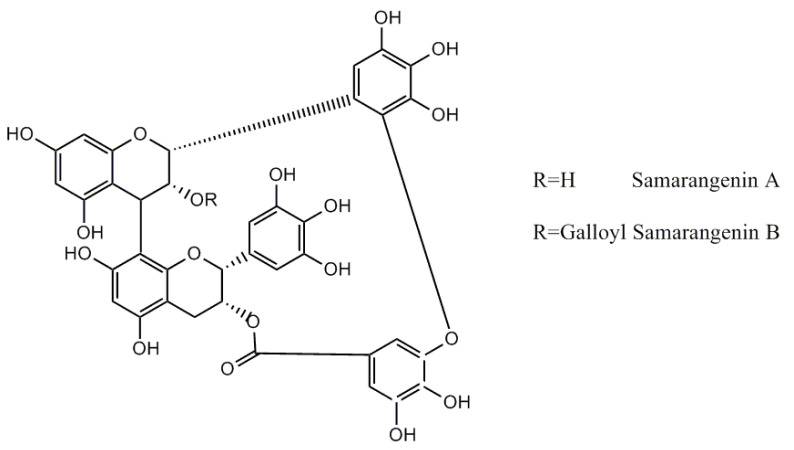
Structure of samarangenins A and B.

**Figure 5 nutrients-13-00165-f005:**
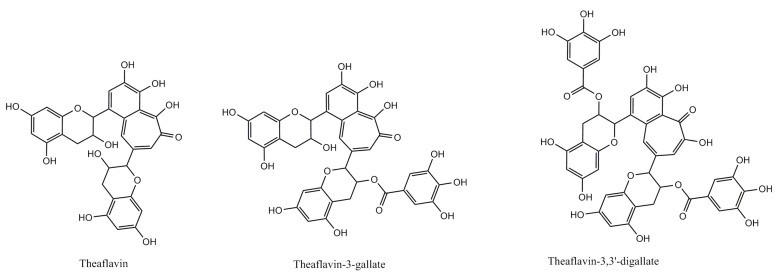
Structure of theaflavins.

**Table 1 nutrients-13-00165-t001:** Influence of proanthocyanidins or flavan-3-ols on periopathogens and their proteolytic enzymes.

Source Plant (If Available)/Active Compound/Extract/Fraction	Periopathogen and Its Proteinase/Toxin	Results	Authors, Year	Ref.
Catechin	*P. gingivalis*	Catechin did not influence the growth of *P. gingivalis* at the concentration tested (20, 40, or 60 µM).	(Lee et al. 2020)	[19]
*Pelargonium sidoides* DC./root extract (PSRE) and proanthocyanidin fraction from PSRE (PACN)	*A. actinomycetemcomitans*	(1) PSRE and PACN significantly reduced bacterial metabolic activity in comparison to the untreated control: (2) 80 µg/mL PSRE, decreased by 57%; (3) 80 µg/mL PACN. decreased by 99%. (4) PSRE and PACN at 100 µg/mL were effective in protecting human gingival fibroblasts from *A. actinomycetemcomitans* infection; (5) PSRE and PACN protected rat gingival fibroblasts from bacterial LPS-induced necrosis.	(Jekabsone et al. 2019)	[20]
The buds of *Castanopsis lamontii* Hance/water extract (CLE) rich in epicatechin (EC) and procyanidin B2 (PB2).	*P. gingivalis*	MICs of CLE, EC, and PB2 against *P. gingivalis* were 0.625, 1.25, and >1.25 mg/mL, respectively.	(Gao et al. 2019)	[21]
c Cranberry fruit (*Vaccinium macrocarpon* Aiton)/proanthocyanidins (PACs) isolated therefrom	*A. actinomycetemcomitans*, leukotoxin	PACs dose-dependently reduced leukotoxin gene expression (ltxB and ltxC, but not ltxA and ltxD) in the two strains of *A. actinomycetemcomitans* tested.	(Amel Ben Lagha et al. 2019)	[22]
Highbush blueberry (*Vaccinium corymbosum* L.) proanthocyanidins (PACs)	*A. actinomycetemcomitans*	At a concentration of 500 μg/mL, the PACs reduced the growth of *A. actinomycetemcomitans* by 62.5% The PACs at concentrations ranging from 500 to 3.9 μg/mL significantly and dose-dependently reduced biofilm formation. More specifically, 31.25 μg/mL of the PACs reduced the growth of bacteria by 23.83% and inhibited biofilm formation by 93.98%. PACs revealed a capacity to reduce biofilm viability, but not biofilm desorption at 500 μg/mL. PACs reduced LtxA cytotoxic towards macrophage-like cells by 100%, 95.4%, and 69.70%, at 125, 62.5, and 31.25 μg/mL, respectively. The PACs protected the oral keratinocytes barrier integrity from damage caused by *A. actinomycetemcomitans*.	(Amel Ben Lagha et al. 2018)	[23]
*Camellia sinensis* (L.) Kuntze/the commercial green tea extract polyphenol content of 98.42%, including 47.92% of (−)-epigallocatechin gallate (EGCG).	*P. gingivalis*, Arg-gingipain, Lys-gingipain	62.5 μg/mL of both the green tea extract and EGCG inhibited the degradation of type I collagen by a *P. gingivalis* culture supernatant by 91.1% and 94.5%, respectively. The green tea extract caused more significant inhibitions of both Arg-gingipain and Lys-gingipain activities than EGCG. More specifically, 125 μg/mL of the green tea extract and EGCG reduced Arg-gingipain activity by 61.82% and 16.46%, while Lys-gingipain activity was reduced by 51.28% and 7.97%, respectively. Green tea extract and EGCG enhanced the barrier function of a gingival keratinocyte model and exerted a protective effect against invasion by *P. gingivalis.*	(Amel Ben Lagha et al. 2018)	[24]
*Pelargonium sidoides* DC./root extract (PSRE) composed mainly of various catechins and prodelphinidin oligomers; and proanthocyanidin fraction from PSRE (PACN) composed of prodelphinidin oligomers.	*P. gingivalis*	PSRE extract significantly reduced the viability of *P. gingivalis*, as well as *S. salivarius* in a dose-dependent manner, starting from the lowest tested concentration-0.02 g/mL (viability reduction by about 74% and 59%, respectively). After treatment with PACN, the *P. gingivalis* viability was not significantly decreased until using 0.05 g/mL. However, starting from this concentration the inhibition of *P. gingivalis* viability was stronger for PACN than for PSRE. Moreover, PACN was less effective against the commensal *S. salivarius* viability.	(Savickiene et al. 2018)	[25]
*Camellia sinensis* (L.) Kuntze Mixture of theaflavins (TFs) from black tea (theaflavin-3-gallate, theaflavin-3′-gallate, and theaflavin-3-3′-digallate, with more than 80% purity)	*P. gingivalis*	TFs dose-dependently inhibited the expression of genes: *fimA*, *hagA*, *rgpA*, and *kgp.* More specifically, TFs at 50 μg/mL inhibited the expression of *fimA*, *hagA*, *rgpA*, and *kgp* by 42%, 47%, 52%, and 53%, respectively. TFs also dose-dependently inhibited the adherence of *P. gingivalis* to keratinocytes and Matrigel^TM^ (250 μg/mL of TFs inhibited adhesion by 73.9% and 64%, respectively). A treatment of gingival keratinocytes with TFs (31.25–125 μg/mL) significantly enhanced tight junction integrity and prevented *P. gingivalis*-mediated tight junction damage, as well as bacterial invasion.	(A Ben Lagha and Grenier 2017)	[26]
*Limonium brasiliense* (Boiss.) Kuntze/70% acetone extract from rhizomes, rich in proanthocyanidins, gallic acid, and epigallocatechin-3-O-gallate/high amount of untypical double linked proanthocyanidins named samarangenins A and B	*P. gingivalis*, Arg-gingipain	*Limonium brasiliense* rhizomes (LBEs) at 100 μg/mL reduced the adhesion of *P. gingivalis* to the human epithelial KB cells by about 80% and at 20 μg/mL reduced the proteolytic activity of the arginin-specific Rgp gingipain by about 75%.	(de Oliveira Caleare et al. 2017)	[27]
*Camellia sinensis* (L.) Kuntze/the commercial green tea extract with polyphenol content of 98.42%, including 47.92% of (−)-epigallocatechin gallate (EGCG).	*P. gingivalis;* expression of several *P. gingivalis* genes involved in host colonization (*fimA*, *hagA*, *hagB*), tissue destruction (*rgpA*, *kgp*), heme acquisition (*hem*), and stress response (*htrA*) investigated	The MIC values of the green tea extract ranged from 250 to 1000 μg/mL, while for EGCG, they ranged from 125 to 500 μg/mL. Synergistic antibacterial effects were observed for the green tea extract or EGCG in combination with metronidazole. The combination of the green tea extract or EGCG and tetracycline resulted mostly in an additive effect. Both substances caused a dose-dependent inhibition of bacterial adherence to oral epithelial cells. Green tea extract and EGCG dose-dependently inhibited the expression of *fimA*, *hagA*, *hag*, *rgpA*, *kgp*, and *hem*. However, both compounds increased the expression of the stress protein *htrA* gene. Green tea extract and EGCG inhibited quorum sensing.	(Fournier-Larente, Morin, and Grenier 2016)	[28]
Persimmon (*Diospyros kaki* Thunb.)/fruit extract (PS-M) contained 21.5 wt% of condensed tannin (proanthocyanidins).	Oral polymicrobial biofilms	The colony forming units (CFUs) were lower in all PS-M and CHX (chlorhexidine) groups compared to the control group. PS-M exerted a dose-dependent effect. PS-M at a dose of 4.0 wt% had the same effect as 0.2 wt% CHX. SEM revealed that the biofilm structures were considerably destroyed in the 4.0 wt% PS-M and 0.2 wt% CHX.	(Tomiyama et al. 2016)	[29]
*Rumex acetosa* L/70% acetone extract from aerial parts., after removal of lipophilic compounds (RA1); (1) epicatechin, (2) catechin, (3) epigallocatechin, (4) gallocatechin, (5) epicatechin-3-O-gallate, (6) epigallocatechin-3-O-gallate, (7) procyanidin B2, (8) procyanidin B2-di-gallate, (9) epicatechin-(4β→6)-epicatechin-3-O-gallate, (10) epicatechin-3-O-gallate-(4β→6)-epicatechin-3-O-gallate, (11) epicatechin-(4β→8)-epicatechin-(4β→8)-catechin, (12) epicatechin-3-O-gallate-(4β→8)-epicatechin-3-O-gallate-(4β→8)-epicatechin-3-O-gallate, (13) epiafzelechin-3-O-gallate-(4β→8)-epicatechin-3-O-gallate, (14) cinnamtannin B1, (15) quercetin-3-O-glucuronide	*P. gingivalis*, Arg-gingipain, Lys-gingipain	RA1 (5 to 15 μg/mL) reduced *P. gingivalis* adhesion to KB cells in a dose-dependent manner to about 90%. Galloylated flawan-3-ols and proanthocyanidins were confirmed to be responsible for this antiadhesive effect with (8) procyanidin B2-di-gallate being the lead compound. Ungalloylated flavan-3-ols and oligomeric proanthocyanidins (1,2,3,4,7,11) were inactive. RA1 and the galloylated proanthocyanidins (5,6,8,9,10,12,13) strongly interacted with the bacterial virulence factor Arg-gingipain, while the corresponding Lys-gingipain was hardly influenced. RA1 does not influence the gene expression of rgpA, kgp, and fimA. RA1 inhibited hemagglutination. In silico docking studies indicated that (8) procyanidin B2-di-gallate interacts with the active side of Arg-gingipain and hemagglutinin from *P. gingivalis*, and the galloylation of the molecule seems to be responsible for the fixation of the ligand to the protein.	(Schmuch et al. 2015)	[30]
*Camellia sinensis* (L.) Kuntze/the commercial black tea extract (with theaflavin content of 40.23%); theaflavin (TF), theaflavin-3,3′-digallate (TFg),	*P. gingivalis*, *Prevotella intermedia*, *Fusobacterium nucleatum*, *A. actinomycetemcomitans*	MIC/MBC values (μg/mL) of black tea, TF, and TFg for *P. gingivalis* and *P. intermedia* were very similar, 500/1000, 125/500, and 250/500, respectively, and significantly higher for *F. nucleatum*, 2000/4000, 250/>1000, and 250/>1000, and *A. Actinomycetemcomitans*, 2000/8000, 250/>1000, and 500/1000. The black tea extract, theaflavin, and theaflavin-3,3′-digallate can potentiate the antibacterial effect of metronidazole and tetracycline against *P. gingivalis*.	(Telma Blanca Lombardo Bedran et al. 2015)	[31]
*Vaccinium angustifolium* Ait./70% ethanolic blueberry extract (phenolic acids, flavonoids, and procyanidins made up 16.6, 12.9, and 2.7% of the blueberry extract, respectively.	*Fusobacterium nucleatum*	The MIC of the blueberry extract against *F. nucleatum* was 1 mg/mL. This concentration also corresponded to the MBC. It was suggested that this property may result from the ability of blueberry polyphenols to chelate iron. Moreover, the blueberry extract at 62.5 μg/mL inhibited *F. nucleatum* biofilm formation by 87.5%.	(Amel Ben Lagha et al. 2015)	[32]
Epigallocatechin gallate (EGCG).	*P. gingivalis*	The MIC of EGCG was 500 μg/mL. EGCG at 500 μg/mL or 5 mg/mL significantly destroyed established *P. gingivalis* biofilms (in these concentrations of EGCG, adenosine triphosphate (ATP) levels were about 40% lower compared to the control). Damage of the cell membranes of *P. gingivalis* were frequently observed in these high concentrations. Moreover, EGCG at sub-MIC levels (10 μg/mL and 100 μg/mL) significantly inhibited *P. gingivalis* biofilm formation (ATP levels were more than 60% lower compared to the control); however, it did not damage the cytoplasmic membrane of *P. gingivalis.*	(Asahi et al. 2014)	[33]
Cranberry *Vaccinium macrocarpon* Ait/non-dialyzable material (NDM) prepared from concentrated juice, rich in proanthocyanidins.	*P. gingivalis* and *F. nucleatum* mixed infection	NDM inhibited coaggregation between *P. gingivalis* and *F. nucleatum* in a dose-dependent manner (starting from 1 mg/mL). NDM inhibited *P. gingivalis* and *F. nucleatum* adhesion to human epithelial cells. The 4 mg/mL of NDM fully inhibited the adhesion of *F. nucleatum* and *P. gingivalis* onto the epithelial cells, leaving the cells entirely free of bacteria.	(Polak et al. 2013)	[34]
*Vitis vinifera* L./commercial proanthocyanidins from grapeseed extract (Leucoselect ^®^, Indena, Italy) combined with H_2_O_2_ and photo-irradiation.	*P. gingivalis*, *S. mutans*	A hydrogen peroxide photolysis system in combination with proanthocyanidin from grapeseed extract synergistically induced damage in *P. gingivalis* and *S. mutans*, leading to killing of these bacteria.	(Ikai et al. 2013)	[35]
Epigallocatechin gallate (EGCG).	*A. actinomycetemcomitans*	Antimicrobial activity was observed at >0.5 mg/mL of EGCG. Alpha-amylase reduced the antimicrobial activity of EGCG, and EGCG inhibited the activity of alpha-amylase. The reason was precipitated alpha-amylase by EGCG after adding to saliva.	(Hara et al. 2012)	[16]
*Vaccinium macrocarpon* Ait./A-type cranberry proanthocyanidins (APAC) and licochalcone A (LA)-chalcone, not proanthocyanidin.	*P. gingivalis*	APAC, at the highest concentration tested (50 μg/mL), did not affect the growth of *P. gingivalis*, whereas licochalcone A completely prevented growth at 10 μg/mL. When the two compounds were used in combination, *P. gingivalis* growth was inhibited in a synergistic manner. On the contrary, licochalcone A had no effect on the adherence of *P. gingivalis* to epithelial cells, but 50 μg/mL of APACs reduced bacterial adherence by approximately 25%. When used in combination, they acted in synergy to inhibit the adherence of *P. gingivalis* to oral epithelial cells. APACs at 25 μg/mL inhibited *P. gingivalis* collagenase by 66%.	(Feldman and Grenier 2012)	[36]
*Myrothamnus flabellifolia* Welw. (MF)/50% EtOH extract, rich in flavan-3-ols and oligomeric proanthocyanidins; epicatechin (EC), epigallocatechin (EGC), gallocatechin (GC).	*P. gingivalis*, Arg-gingipain, Lys-gingipain	MF dose-dependently (0.1–1.0 mg/mL) inhibited *P. gingivalis* epithelial cell attachment or invasion (by about 50% at 1 mg/mL); however, bacterial growth was not influenced. Reduced adhesion was observed after pre-treatment of bacteria, pre-treatment of KB cells, as well as co-incubation of bacteria together with KB cells in the presence of MF (0.1 mg/mL). The MF extract (1–1000 μg/mL) showed inhibition of bacterial haemagglutinin. The MF extract at 50 μg/mL reduced Arg-gingipain by 70–80% and also inhibited Lys-gingipain, but to a lesser extent. Fimbrillin (*fimA*) and Arg-gingipain (*rgpA*), but not Lys-gingipain (*kgp*), encoding genes were upregulated by 10 and 100 μg/mL of MF. EGC and GC at 3 mM reduced the *P. gingivalis* adhesion to KB cells by about 40%. EC, EGC, and GC inhibited hemagglutination in a dose-dependent manner (30–300 μM). A reduction of proanthocyanidin titers in the bacteria-free supernatant by about 40% after incubation *P. gingivalis* with proanthocyanidins was observed.	(Löhr et al. 2011)	[37]
Catechins: (+)-catechin (C), (−)-epicatechin (EC), (−)-gallocatechin (GC), (−)-epigallocatechin (EGC), (−)-catechin gallate (CG), (−)-epicatechin gallate (ECG), (−)-gallocatechin gallate (GCG), (−)-epigallocatechin gallate (EGCG).	*Eikenella corrodens*	1 mM of GC, EGC, CG, ECG, GCG, and EGCG significantly inhibited *E. corrodens* biofilm formation, whereas EC and C had no effect. Moreover, the catechins with the galloyl group (CG, ECG, GCG, EGCG) remarkably inhibited biofilm formation even at 0.1 mM, whereas the effects of catechins with only the pyrogallol-type B-ring (GC, EGC) were weaker, starting from 0.5 mM and 0.25 mM, respectively. Only catechins with the galloyl group revealed bactericidal activity at a 1 mM concentration; however, none of the catechins showed bactericidal activities at a 0.1 mM concentration, which suggests that catechins with the pyrogallol-type B-ring and/or the galloyl group inhibit biofilm formation at sub-MIC, by a mechanism other than bactericidal activity. As the research showed, this may by through the interference with the AI-2-mediated QS system in *E. corrodens*.	(Matsunaga et al. 2010)	[38]
Cranberry fruit (*Vaccinium macrocarpon* Ait.)/isolated A-type cranberry proanthocyanidins (AC-PACs),	*P. gingivalis*	AC-PACs inhibited biofilm formation by 45% and 60% at concentrations of 50 and 100 μg/mL and inhibited *P. gingivalis* adherence to epithelial cells by 37.5% and 54.1%, respectively. At these concentrations, AC-PACs also inhibited the adherence of *P. gingivalis* to Matrigel-coated polystyrene surfaces. The 25, 50, and 100 μg/mL of AC-PACs inhibited type I collagen degradation by a *P. gingivalis* culture supernatant by about 50%, 74%, and 89%, respectively. At all the concentrations tested (25–100 μg/mL), AC-PACs did not significantly affect the growth of *P. gingivalis*.	(Vu Dang La, Howell, and Grenier 2010)	[39]
Cranberry *Vaccinium macrocarpon* Ait/non-dialyzable material (NDM) prepared from concentrated juice, containing 65.1% proanthocyanidins.	*Peptostreptococcus micros*	Treatment of monocyte-derived macrophages, as well as oral epithelial cells with the cell wall of *P. micros* decreased their cell viability; however, adding the cranberry fraction (25–50 μg/mL) prior to treating cells with the *P. micros* cell wall dose-dependently protected these cell lines from the toxic effect.	(Vu Dang La, Labrecque, and Grenier 2009)	[40]
Same as above.	*P. gingivalis*	NDM significantly prevented the attachment of *P. gingivalis* to surfaces coated with type I collagen, fibrinogen, or human serum. NDM inhibited the biofilm formation of *P. gingivalis*; however, it had no effect on the growth and viability of bacteria.	(Labrecque et al. 2006)	[41]
Same as above.	Arg-gingipain, Lys-gingipain, dipeptidyl peptidase IV of *P. gingivalis*; trypsin-like protease of *T. forsythia*; chymotrypsin- like protease of *T. denticola*	NDM dose-dependently inhibited the proteinases of *P. gingivalis*, *T. forsythia*, and *T. denticola* (10–150 μg/mL); however, the trypsin-like activity of *T. forsythia* was only slightly sensitive to NDM. 50 μg/mL of NDM significantly reduced the collagenase activity of *P. gingivalis* (by 30%) and the capability of *P. gingivalis* to degrade transferrin (by about 20%). Degradation of type I collagen and transferrin by *P. gingivalis* was completely or almost completely inhibited by 100 μg/mL and 150 μg/mL of NDM, respectively.	(Charles Bodet et al. 2006)	[42]
Apple (*Malus* domestica L.)/apple fraction (AP) rich in proanthocyanidins; apple condensed tannin (ACT) isolated from AP; hops (*Humulus japonicus* Siebold & Zucc.)/hop bract polyphenol (HBP) fraction rich in proanthocyanidins; HMW-HBP (high molecular weight fraction) and LMW-HBP (low molecular weight fraction) separated from HBP; HMW-HBP mainly containing 8 to 22 mer proanthocyanidins; EGCG: (−)-epigallocatechin gallate.	*P. gingivalis*, Arg- and Lys-gingipains	None of the fractions revealed bactericidal activity or suppression of bacterial growth at concentrations of 1 and 10 μg/mL. The studied fractions at 10 μg/mL significantly protected PDL (periodontal ligament) cells’ viability from the effect of *P. gingivalis* infection, although EGCG and LMW-HBT showed slightly lower effects than the others. Even at 1 μg/mL, AP, ACT, HBP, and HMW-HBP demonstrated protective effects. All of the fractions revealed significant inhibitory effects toward the proteolytic activities of Rgp and Kgp in a dose-dependent manner, with the ratios ranging from 70% to 95% at 10 and100 μg/mL. At lower doses (0.1 and 1 μg/mL), EGCG showed the greatest effect, followed by ACT and AP.	(Inaba et al. 2005)	[43]
(−)-Epigallocatechin gallate (EGCG), epicatechin gallate (ECG), epigallocatechin (EGC), epicatechin (EC), (−)-gallocatechin gallate (GCG), catechin gallate (CG), gallocatechin (GC), (−)-catechin (C), gallic acid (G).	*P. gingivalis*, Arg- and Lys-gingipains	Catechin derivatives, containing the galloyl moiety, which includes EGCG, ECG, GCG, and CG, significantly inhibited the Arg-gingipains’ activity. The IC_50_s ranged from 3 to 5 mM. Non-galloylated catechins, EGC and GC, moderately inhibited Arg-gingipains’ activity (IC_50_s, 20 mM), while EC, C, and G were not effective, with IC_50_ greater than 300 mM. Furthermore, some of the catechin derivatives (galloylated) also inhibited the Lys-gingipains’ activity, though to a lesser extent than the inhibition of the Arg-gingipains’ activity.	(Okamoto et al. 2004)	[44]
*Camellia sinensis* (L.) Kuntze/Tea polyphenol mixture (TP), (+) catechin (C), (−) epicatechin (EC), (+) gallocatechin (GC), (−) epigallocatechin (EGC), (−) epicatechin gallate (ECG), (−) epigallocatechin gallate (EGCG), (−) gallocatechin gallate (GCG).	Short-chain fatty acid (n-butyric and propionic acid), as well as phenylacetic acid production by *P. gingivalis*	The production of n-butyric and propionic acid in general anaerobic medium (GAM) was inhibited by TP in a dose-dependent manner; complete inhibition was seen at a concentration of 1.0–2.0 mg/mL. EGCG, a major component of tea polyphenols, inhibited the production of phenylacetic acid at 0.5 mg/mL. EGCG and other galloylated catechins, ECG and GCG, inhibited the reaction leading to the production of phenylacetic acid from L-phenylalanine and phenylpyruvic acid. However, C, GC, EC, and EGC did not inhibit those reactions. Moreover, the growth of *P. gingivalis* was inhibited by EGCG (strong at 0.5 mg/mL).	(Senji Sakanaka and Okada 2004)	[45]
Elm *(Ulmus macrocarpa* Hance)/extract (EE) (n-butanol fraction from extract of *Ulmi cortex*) containing 20% of procyanidins and the mixture of procyanidin oligomers (PO).	trypsin-like enzymes from *T. denticola* and *P. gingivalis*	Both EE and PO (0.1–0.01%) effectively inhibited the activity of the *T. denticola* proteases, whereas EE inhibited *P. gingivalis* proteases’ activity less than PO. PO, at a concentration of 0.1–0.01%, reduced the trypsin-like enzymes of *T. denticola* to 34–58% activity and the trypsin-like enzymes of *P. gingivalis* to 39–73% activity, whereas the same concentrations of the elm extract reduced the *T. denticola* enzyme activity to 40–89% and *P. gingivalis* to 49–91%.	(Song et al. 2003)	[46]
*Camellia sinensis* (L.) Kuntze/the green tea catechin well-purified by Sunphenon ^®^ (Taiyo Kagaku, Yokkaichi, Mie, Japan) prepared from Japanese green tea; details about the composition the of extract not provided.	*P. gingivalis*, *Prevotella* species	The MICs of the green tea catechin for *P. gingivalis*, *P. intermedia*, and *P. nigrescens* were 1.0 mg/mL. The green tea catechin showed bactericidal effects against all three bacteria. However, a high concentration of catechin was used (4 mg/mL).	(Hirasawa et al. 2002)	[47]
*Camellia sinensis* (L.) Kuntze/tea polyphenol mixture (TP) (+) catechin (C), (−) epicatechin (EC), (+) gallocatechin (GC), (−) epigallocatechin (EGC), (−) epicatechin gallate (ECG), (−) epigallocatechin gallate (EGCG), (−) gallocatechin gallate (GCG).	*P. gingivalis*	EGCG completely inhibited the growth of three strains of *P. gingivalis* at concentrations of 250 or 500 μg/mL. The MICs for other polyphenols were 1000 μg/mL. TP at the concentration of 100 μg/mL reduced the adherence of *P. gingivalis* to human buccal epithelial cells by about 70%. All of the compounds inhibited the adherence of *P. gingivalis* to epithelial cells. However, the inhibitory effect was pronounced with catechin derivatives having a galloyl moiety: EGCG, GCG, and ECG (at 250 μg/mL, they almost completely inhibited the adherence of *P. gingivalis* to epithelial cells). Even at 7.8 μg/mL, EGCG or ECG reduced the adhered bacterial cells by about 70%. Inhibition of the adherence of *P. gingivalis* to epithelial cells was much more effective when EGCG was preincubated with bacteria than with epithelial cells.	(Sakanaka et al. 1996)	[48]

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
