# Peer review of "Proanthocyanidins and Flavan-3-Ols in the Prevention and Treatment of Periodontitis—Antibacterial Effects"

_nutrients, 2021, doi:10.3390/nu13010165_

Round 1

Reviewer 1 Report

This manuscript is written on a clinically and biologically relevant topic.  The manuscript is well written with only moderate grammatical correction necessary.  

Minor comments:

Line 206: change "reach" to "rich".

Line 207:  "count to the" is unclear.

Author Response

The Authors' response is in the attached PDF file.

Reviewer 2 Report

Thank you very much for the opportunity of reviewing this paper. The topic of this systematic review is very interesting. Hovewer, it has important limitations.

The paper is not justify properly. There are some previous reviews excluded during the selection. It is suggested to describe what is new about conducting this review.

Another concern is the use of only one database to run the search strategy. The guidelines to perform a review indicate to use, at least, two databases. Even, last recommendations point the use of four databases (Muta et al., European Journal of Epidemiology. 2020. 35:49–60). This limitation could have led to selection bias.

It was not defined the search question and it difficult the evaluation of the search strategy. Moreover, the search strategy did not include synonyms as “gingival disease”, e.g.

The evaluation of the quality of the included studies was not included. The quality and the assessment of the bias of the studies could have influence in the interpretation of the results of the paper and its conclusions.

Author Response

The authors' response is in the attached PDF file.

Reviewer 3 Report

This review manuscript describes about the abilities of proanthocyanidins and flavan-3-ols from several foodstuffs and medicinal herbs on periodontopathic bacteria and periodontal tissues.  Overall, this review is a well-written, very interesting and relatively-good article.

This reviewer recommends the authors to rearrange and reconstruct the “Results and Discussion” section based on the functions of proanthocyanidins and flavan-3-ols, such as “antibacterial effects on periopathogens”, “inflammation modulatory effects on host cells and tissues”, “Influence on periodontitis in animal model” and “Clinical trials”, as shown in Figure 3.

These descriptions should make this review article great and easy to understand.

Specific comments:

Line 47: Please change “Actinobacillus” to “Aggregatibacter”.  Please check this issue through the manuscript.

Line 64: Please change “MMPs” to “(MMPs)”.

Line 68: Please change “matrix metalloproteinases (MMP)” to “MMP”.

Line 141: What does “(3)” mean?

Figure 3: Please change “Periopathogenes” to “periopathogens”.

Table 1: Please change “proteinase/toxine” to “proteinase/toxin”.

Table 1 in Page 7: Please change “62,5” to “62.5”.  “,” to “.”.  Please check this issue through the manuscript.

Table 1 in Page 10: Please “H2O2” to “H2O2”.  “2 (subscript)” Please check this issue through the manuscript.

Table 1 in Page 11: Please change “abaout” to “about”.

Table 1 in Page 12: Please change “depen- dent” to “dependent”.

Table 1 in Page 12: Please insert “,” between “GCg” and “Cg”.

Lines 184-187: Please change to “P. gingivalis”, “A. actinomycetemcomitans”, “F. nucleatum”, “P. intermedia”, “T. denticola”, “T. forsythia” and “Peptostreptococcus micros”.

Line 222: Please spell out “EPR”.  “Electron Paramagnetic Resonance”?

Line 235: Please create a space between “10-150” and “μg/ml”.

Line 251: Please change “Lohr” to “Löhr”.

Line 252: Please change “EtOH” to “ethanol”.

Line 267: Please change “alos” to “also”.

Line 307: Please insert “have shown” between “Sakanaka et al.” and “the same extract”.

Line 316: Please change “At sub-MIC EGCG,” to “At sub-MIC, EGCG”.

Line 317: Please insert “.” just after “P. gingivalis”.

Line 319: Please insert “.” just after “Gao et al”.

Line 326: Please change “comensal” to “commensal”.

Line 352: Please change “PCAs” to “PACs”.

Line 375: Please create a space between “thirty” and “reported”.

Lines 376-378: Please change to “P. gingivalis”, “A. actinomycetemcomitans”, “F. nucleatum”, “P. intermedia”, “T. denticola”, “T. forsythia” and “P. micros”.

Line 380: Please create a space between “P.” and “gingivalis”.

Author Response

(The authors gave the same response as above.)

Reviewer 4 Report

The authors summarised all the available evidence on he effects of proanthocyanidins and Flavan-3-ols on pathogens associated with periodontitis and their efforts must be commended.

  • Line 17: bioactivities may not be the appropriate work here
  • Line 67: please replace “object” with “molecule”
  • In table 1, column one: you may want to present it in the following way if possible

Product/plant (active molecule)

e.g. Green tea (polyphenols)

  • Line 375: thirty reported
  • I would like to see subheadings inside the text that will facilitate the reader understand where are the in vitro studies, animal studies and clinical studies summarized respectively.
  • Please elaborate on the clinical applications and the evidence we have available today also make reference to this aspect in the abstract

Author Response

(The authors gave the same response as above.)

Round 2

Reviewer 2 Report

Thank you for the reply.

- The articles finally selected sum 81 (n=31, n= 36, n= 11, n=3), not 65.

- The authors indicate in the reply that they have filtered out the low-quality reports during the preliminary assessment (sections 2.2 and 2.3) but this information does not appear in these sections.

Author Response

- The articles finally selected sum 81 (n=31, n= 36, n= 11, n=3), not 65.

1. We greatly appreciete the Reviewer's  remark on the apparently mismatching reference numbers. Indeed, it needs a straightforward clarification that some of cited studies appeared in both topics, i.e in antibacterial and antiinflammatory. There were 16 such papers and it has been explicitly stated in the main text (section 3. introductory paragraph) and in the caption to Figure 3 (the flowchart).

The added explanations read: "In some references (n = 16), the in vitro antibacterial tests were performed concurrently with investigations into the immunomodulatory effects in host cells or in animal models. Results of these studies pertaining to the inflammatory processes are reviewed in our other paper, separately [18]"

We have also modified the flowchart to make it less ambiguous in this respect and better reflect the used search logic.

- The authors indicate in the reply that they have filtered out the low-quality reports during the preliminary assessment (sections 2.2 and 2.3) but this information does not appear in these sections.

2. Indeed, we've ommitted this explanation. However, as suggested by the Reviewer, it has been now added at the end of section 2.2.

Reviewer 3 Report

This revised paper is considerably improved. 

Author Response

Thank you very much for your previous comments and suggestions. We greatly appreciate the effort and the positive opinion about the revised manuscript.